

# A new model for freedom of movement using connectomic analysis

Diego Alonzo Rodríguez-Méndez[1], Daniel San-Juan[2], Mark Hallett[3], Chris G. Antonopoulos[4], Erick López-Reynoso[1] and Ricardo Lara-Ramírez[5]

[1] Facultad de Ciencias, Universidad Autónoma del Estado de México, Toluca, Estado de México, México
[2] Epilepsy Clinic, Instituto Nacional de Neurología y Neurocirugía, Mexico City, Mexico
[3] Human Motor Control Section, Medical Neurology Branch, NINDS, National Institutes of Health, Bethesda, MD, United States of America
[4] Department of Mathematical Sciences, University of Essex, Wivenhoe Park, United Kingdom
[5] Centro de Investigación en Ciencias Biológicas Aplicadas, Universidad Autónoma del Estado de México, Toluca, Estado de México, México

Corresponding author
Ricardo Lara-Ramírez,
rlarar@uaemex.mx

## ABSTRACT

The problem of whether we can execute free acts or not is central in philosophical thought, and it has been studied by numerous scholars throughout the centuries. Recently, neurosciences have entered this topic contributing new data and insights into the neuroanatomical basis of cognitive processes. With the advent of connectomics, a more refined landscape of brain connectivity can be analysed at an unprecedented level of detail. Here, we identify the connectivity network involved in the movement process from a connectomics point of view, from its motivation through its execution until the sense of agency develops. We constructed a "volitional network" using data derived from the Brainnetome Atlas database considering areas involved in volitional processes as known in the literature. We divided this process into eight processes and used Graph Theory to measure several structural properties of the network. Our results show that the volitional network is small-world and that it contains four communities. Nodes of the right hemisphere are contained in three of these communities whereas nodes of the left hemisphere only in two. Centrality measures indicate the nucleus accumbens is one of the most connected nodes in the network. Extensive connectivity is observed in all processes except in Decision (to move) and modulation of Agency, which might correlate with a mismatch mechanism for perception of Agency.

# INTRODUCTION

Freedom of action is fundamental to our self-awareness as humans. Volition is generally believed to be the causative force of many of our actions. This concept includes, for example, the notion of free will for movement. We feel capable of deciding and executing our actions and, having made them, we feel the authors of these actions. The sense of being the author is called Agency, that is, we feel aware of having done a movement by our own decision (*Nahab et al., 2011*; *Pacherie, 2013*; *Haggard, 2017*). However, whether we are free to choose our own decisions and our movements is debatable

(*Van Inwagen, 1975*; *Libet, 1999*; *Harris, 2012*). *Hallett (2009)*, *Hallett (2016)*, and *Kranick & Hallett (2013)* suggested a series of steps to model the process of voluntary movement and mapped them onto neuroanatomical regions of the cerebral cortex. From this, we can discern several processes needed to accomplish voluntary movement and the specific brain areas regulating each of them as identified from the literature (Table 1).

Motivation is where internal and external influences are received that initiate voluntary activity (*Hallett, 2007*). Internal influences involve limbic areas such as the amygdala, nucleus accumbens, and the orbitofrontal cortex with final integration in the anterior cingulate cortex (ACC), while external influences are processed in the parietal lobe (*Hallett, 2007*). In an fMRI study, Lee and collaborators (*2012*) found greater activation of the insular cortex (which also connects to ACC) when participants decided to act for intrinsic reasons versus when they decided to act for extrinsic reasons, which activated the posterior cingulate cortex (PCC). These experiments suggested that the PCC could act as a modulator of motivation (*Sumner et al., 2007*; *Lee et al., 2012*).

Another process, called Planning, is where intention is formed in the premotor areas and aids in the decision between the broad spectrum of movements that would fulfil motivation (*Deiber et al., 1991*; *Deiber et al., 1996*). Using positron emission tomography (PET), *Deiber et al. (1991)* showed that there is a bilateral increase of activity in the premotor cortex (supplementary motor area, SMA, and prefrontal BA46/9) and the parietal lobe, particularly in the superior parietal cortex on the lateral and medial surfaces, of subjects allowed to decide what movement to make with respect to when making a pre-determined movement. In a subsequent experiment, Deiber and collaborators (*1996*) showed that the anterior part of the SMA (pre-SMA) is the main area preferentially activated when a movement is internally chosen (compared to a cued pre-defined condition). Also, greater activation was measured in the prefrontal cortex (BA10), the anterior cingulate cortex (BA32), and the left anterior parietal cortex (BA40) (*Deiber et al., 1996*). These experiments are further supported by clinical observation. Archibald and her team (*2001*) found that dysfunction in structures of the mesial frontal lobe and frontostriatal pathways (SMA, cingulate gyrus, and basal ganglia) are involved in the pathophysiology of utilisation behaviour, grasp reflex, and manual groping behaviour, rendering the subject more prone to act on reflex alone.

Besides selecting what movement to make, we also decide when to make it, that is, to make it immediately or let a certain amount of time pass before the action is made. "Libet's experiment" is the referent for this kind of study (*Libet, Wright & Gleason, 1983*). Libet's experiment confirms that cerebral initiation of voluntary movements begins unconsciously, before there is any subject awareness of a decision which has already been initiated cerebrally (*Libet, Wright & Gleason, 1983*). After Libet et al., more experiments were made to determine which parts of the brain were activated and at which time. Greater activation in the pre-SMA (*Deiber et al., 1999*) and the DLPFC (*Jahanshahi et al., 1995*) was found in free timing arrangements. These studies served as a basis for more recent ones where it was possible to predict which future movement would be made based on the preparatory activity of the pre-SMA and frontal cortex even ten seconds before the willing to move enters awareness (*Soon et al., 2008*).
**Table 1 Network areas organisation.** First column, tasks. Second column, region names; note that pre-SMA is part of both Planning and Timing tasks. Third column, BA codes. Fourth column, Brainnetome Atlas database sub-area names. Fifth column, references. N/A, not applicable.

| Process | Region name | Brodmann area (BA) name | Brainnetome sub-area name | References |
|---------|-------------|-------------------------|----------------------------|------------|
| Motivation | Limbic areas (Amygdala and nucleus accumbens) | N/A | mAmyg_l, mAmyg_r, lAmyg_l, lAmyg_r, NAC_l, NAC_r | |
| | Frontal area (Orbitofrontal cortex) | BA11 | A11l_l, A11l_r, A11m_l, A11m_r | *Hallett (2007)*; *Lee et al. (2012)* |
| | Insular cortex | BA13, BA14 and BA16 | G_l, G_r, vla_l, vla_r, dla_l, dla_r, vId/vIg_l, vId/vIg_r, dIg_l, dIg_r, dId_l, dId_r | |
| Modulation of Motivation | Posterior cingulate cortex | BA31 and BA23 | A31_l, A31_r, A23d_l, A23d_r, A23v_l, A23v_r, A23c_l, A23c_r | *Lee et al. (2012)* |
| Planning | Prefrontal cortex | BA9 and BA10 | A9l_l, A9l_r, A9m_l, A9m_rA10m_l, A10m_r, A10l_l, A10l_r | *Deiber et al. (1991)* |
| | Anterior cingulate cortex | BA24 and BA32 | A24rv_l, A24rv_r, A24cd_l, A24cd_r, A32p_l, A32p_r, A32sg_l, A32sg_r, | *Deiber et al. (1996)* |
| | Left anterior parietal cortex | BA40 | A40rd_l, A40rv_l | *Deiber et al. (1996)* |
| | Pre-supplementary motor area | BA6 | A6dl_l, A6dl_r, A6m_l, A6m_r, A6vl_l, A6vl_r, A6cdl_l, A6cdl_r, A6cvl_l, A6cvl_r | *Deiber et al. (1999)*; *Archibald, Mateer & Kerns (2001)*; *Sumner et al. (2007)*; *Soon et al. (2008)* |
| Timing | Pre-supplementary motor area | BA6 | A6dl_l, A6dl_r, A6m_l, A6m_r, A6vl_l, A6vl_r, A6cdl_l, A6cdl_r, A6cvl_l, A6cvl_r | *Deiber et al. (1999)*; *Soon et al. (2008)* |
| | Dorsolateral prefrontal cortex | BA9 (lateral), BA46 | A9l_l, A9l_r, A9/46d_l, A9/46d_r | *Jahanshahi et al. (1995)*; *Soon et al. (2008)* |
| Decision | Dorsal frontomedial cortex | BA9 (medial) | A9m_l, A9m_r | *Brass & Haggard (2007)* |
| Execution | Primary motor cortex | BA4 | A4hf_l, A4hf_r, A4ul_l, A4ul_r, A4t_l, A4t_r, A4tl_l, A4tl_r, A4ll_l, A4ll_r | *Penfield & Rasmussen (1950)*; *Carlson (2014)* |
| Agency | Insular Cortex | BA13, BA14 and BA16 | G_l, G_r, vla_l, vla_r, dla_l, dla_r, vId/vIg_l, vId/vIg_r, dIg_l, dIg_r, dId_l, dId_r | *Farrer & Frith (2002)*; *Farrer et al. (2003)* |
| Modulation of Agency | Right inferior parietal cortex (Right angular gyrus, right temporoparietal junction) | BA39 | A39c_r, A39rd_r, A39rv_r | *Ruby & Decety (2001)*; *Bzdok et al. (2013)* |
| | Extrastriate body area | BA19 | V5_MT_plus_l, V5_MT_plus_r | *Astafiev et al. (2004)*; *David et al. (2007)* |
| | Caudoposterior superior temporal sulcus | BA22 | cpSTS_l, cpSTS_r | *Nahab et al. (2011)* |

Rodríguez-Méndez et al. (2022), *PeerJ*, DOI 10.7717/peerj.13602

As the brain is along its way to perform an action, it also makes a decision of whether to do it or not. Using EEG, *Parkinson & Haggard (2015)* showed that intentional action and intentional inhibition, when subjects have to choose, are different from their reactive counterparts when subjects are instructed, given that the time in which they are elicited is longer in intentional processes. This additional time might reflect the time to choose between action or inhibition (*Parkinson & Haggard, 2015*). This concept was also researched by *Leocani et al. (2000)* in a go/no-go auditory reaction time experiment, who found that after a no-go tone, bilateral inhibition occurred at a time corresponding to the mean RT of go tones. In another study, *Brass & Haggard (2007)* used fMRI to determine which parts of the brain were more active when internally preparing an action and then restraining the movement compared with when the movement was made. They showed that inhibition of intentional actions is mediated by the dorsal fronto-medial cortex (dFMC, medial BA9), suggesting that inhibition of intentional action involves cortical areas different and upstream from intention generation and execution of action (*Brass & Haggard, 2007*). This may be regarded as a substrate for freedom of movement (we are free to choose what not to do), although care should be taken with this interpretation since voluntary inhibition of movement may also have an unconscious origin (*Astafiev et al., 2004*).

For the execution of movement, the most important cortical region is the primary motor cortex, which is located in the precentral gyrus (*Carlson, 2014*). Various stimulation studies have demonstrated that activation of neurons localised to specific places of the primary motor cortex provokes particular movements of the body (*Carlson, 2014*).

Finally, another process is called Agency, which is the feeling that leads us to attribute an action to ourselves rather than to someone else (*Farrer et al., 2003*). This perception is inferred after an action has been made by "retrospectively" comparing the actual effects of actions against their intended effects (*Chambon, Moore & Haggard, 2015*). When an action does not match the intention, or is not preceded by an intention, a mismatch is detected and gives the sense of an involuntary movement (*Nahab et al., 2011*). This phenomenon is caused by feed-forward signals that come from the premotor and motor cortices to the parietal area which give rise to the sense of willingness (*Hallett, 2016*). Then, feed-forward signals are compared with feedback signals from the movement generated in the parietal area; if there is no mismatch, a perception of agency can arise (*Farrer et al., 2008*; *Kranick & Hallett, 2013*; *Chambon et al., 2013*; *Chambon, Moore & Haggard, 2015*; *Hallett, 2016*; *Haggard, 2017*; *Voss et al., 2017*). Specifically, the initiation of this process is mediated by monitoring the angular gyrus to action selection in the dorsolateral prefrontal cortex (*Chambon et al., 2013*). This phenomenon can be disrupted using transcranial magnetic stimulation over the inferior parietal cortex, preventing the sense of agency (*Chambon, Moore & Haggard, 2015*).

However, studies have questioned a traditional ordinal representation of a sequence of steps to explain cognitive processes and their associated actions, particularly with regards to the onset and offset of each activity (*Viswanathan et al., 2020*) and also from an ecological perspective (*Cisek & Kalaska, 2010*). Thus, an unconstrained representation is needed to better understand the neuroanatomical basis and temporal dynamics of a

particular action such as movement. Also, previous models of the movement process have considered only events preceding movement, such as "the What", "the When" and "the Whether" components (*Zapparoli, Seghezzi & Paulesu, 2017*; *Zapparoli et al., 2018*), and not considered post-movement aspects such as Agency.

Recent advancements in the mapping of white matter tracts have allowed the identification of connectivity among different subregions of the cerebral cortex, which have led to many new insights about brain organisation (*Behrens & Sporns, 2012*). The so-called connectomics approach has proven very useful for the identification (parcellation) of smaller cortical areas communicating with other brain regions forming specific networks across the white matter (*de Reus & and Heuvel, 2013*; *Hinne et al., 2015*). Currently, it is possible to access databases of connectograms for the analysis of specific brain areas and their connections with other cortical regions (*Fan et al., 2016*).

Here, we identify the connectivity network involved in the movement process adopting a connectomics approach, from its motivation through its execution and sense of agency. We constructed a "volitional network" using data derived from the Brainnetome Atlas database considering areas known to be involved in volitional processes as found in the literature. We divided this process into eight processes and used Graph Theory to measure several structural properties of this network which we treated as undirected. Our results show that the volitional network is small-world and that it contains four communities. Nodes of the right hemisphere are contained in three of these communities whereas nodes of the left hemisphere only in two. Centrality measures indicate that the nucleus accumbens is one of the most important nodes in the network. Our analysis provides a new model of the neuronal connectivity regulating voluntary movement with a formal graph theoretical analysis that extends previous knowledge on the neuroanatomical basis of volition.

## MATERIALS & METHODS

### Brain regions

We searched the literature to identify which areas are implicated in the movement-generation process (Table 1). In these empirical experiments, subjects were tested with different movement paradigms to account for distinct properties of volition. Areas thus identified include cortical areas following Brodmann's classification, *i.e.*, Brodmann areas (BAs), and subcortical areas such as the amygdala or nucleus accumbens. After identifying these areas, we tracked their location in the Brainnetome Atlas database (*Fan et al., 2016*) which offers a more refined parcellation of the brain. We treated regions derived from the Brainnetome Atlas database as sub-areas; hence, a given BA can contain one or many sub-areas. All sub-areas identified in the Brainnetome Atlas database are indicated in Table 1. Not all areas reported in the literature involved in the different processes (see below) are found in the Brainnetome Atlas database and were thus excluded from our analysis. This resulted in a total of 82 sub-areas comprising cortical and subcortical areas known from empirical studies found in the literature to be involved in volition (Table 1).

Some clarifications regarding region selection are needed. Extrastriate body area (EBA) activity is present in a way that it is more active when the visual feedback is incongruent

with the subjects' movements (*David et al., 2007*). This area is represented in our model by the MT/V5 cluster for connectivity purposes, as these two areas overlap considerably (*Ferri et al., 2013*). Also, BA6 contains regions from pre-SMA, SMA, and premotor cortex, which show—at least to some extent—different functions (see, for example, *Rahimpour, Rajkumar & Hallett, 2022*). Since the pre-SMA part of BA6 is most relevant for Planning, while the SMA and premotor cortex parts of BA6 are more related to Execution (or at least more to the transition from Planning to Execution) (*Rahimpour, Rajkumar & Hallett, 2022*), in our analysis we consider only the pre-SMA part of BA6 as a node of interest, and thus any discussion on BA6 in this study refers to its pre-SMA part.

## Re-classification of Hallett's steps

We reclassified Hallett's (*2016*) steps for volition to allow for the explicit incorporation of time (when to move) and decision to move (*i.e.*, whether to move or not; for simplicity, we refer to this process as "Decision" only). To avoid a strict order of predefined events, we treated these steps as "processes" which could be active in a given sequence, simultaneously, or have different orders of activation. This reclassification rendered eight processes, namely, Motivation, Modulation of Motivation, Planning (How to move), Timing (When to move), Decision (Whether to move or not), Execution, Agency, and Modulation of Agency. We propose this new classification scheme as opposed to the simpler, three-part "What"/"When"/"Whether" model (*Zapparoli, Seghezzi & Paulesu, 2017*) since it incorporates more processes such as Agency generating a more detailed, realistic, and comprehensive model of the movement process. Regions and areas involved in each process are shown in Fig. 1.

However, we had to make some generalisations since the detail of the Brainnetome Atlas database is not suitable for comparing the whole of frontal lobe's connections to specific pre-motor areas. For example, for the motivational part, we used the amygdala and the nucleus accumbens as areas representative of the limbic system since these areas are closely related to motivation by both appetitive and aversive stimuli (*Fonberg, 1986*; *Salamone, 1994*). We also used the orbitofrontal cortex as representative of the frontal cortex as it may perform aspects of reinforcement value that govern choice behaviour (*Cardinal et al., 2002*).

## Network analysis

With all 82 sub-areas identified and their connections we reconstructed a network that we called "volitional network" and performed a quantitative analysis of its properties and structure using Graph-Theory (*Farahani, Karwowski & Lighthall, 2019*). First, we considered each sub-area identified in the Brainnetome Atlas database as a node and the physical connections among sub-areas as edges in the network. Next, we considered this network as undirected, namely, as a network where connections are bidirectional, meaning a network involving all nodes with no restriction on the sequence of processes.

To ascertain the relative importance of each node in the network we calculated several centrality measures (*Newman, 2010*). These measures use Graph Theory to calculate their "importance" in the network and each has its own definition of "importance". They are

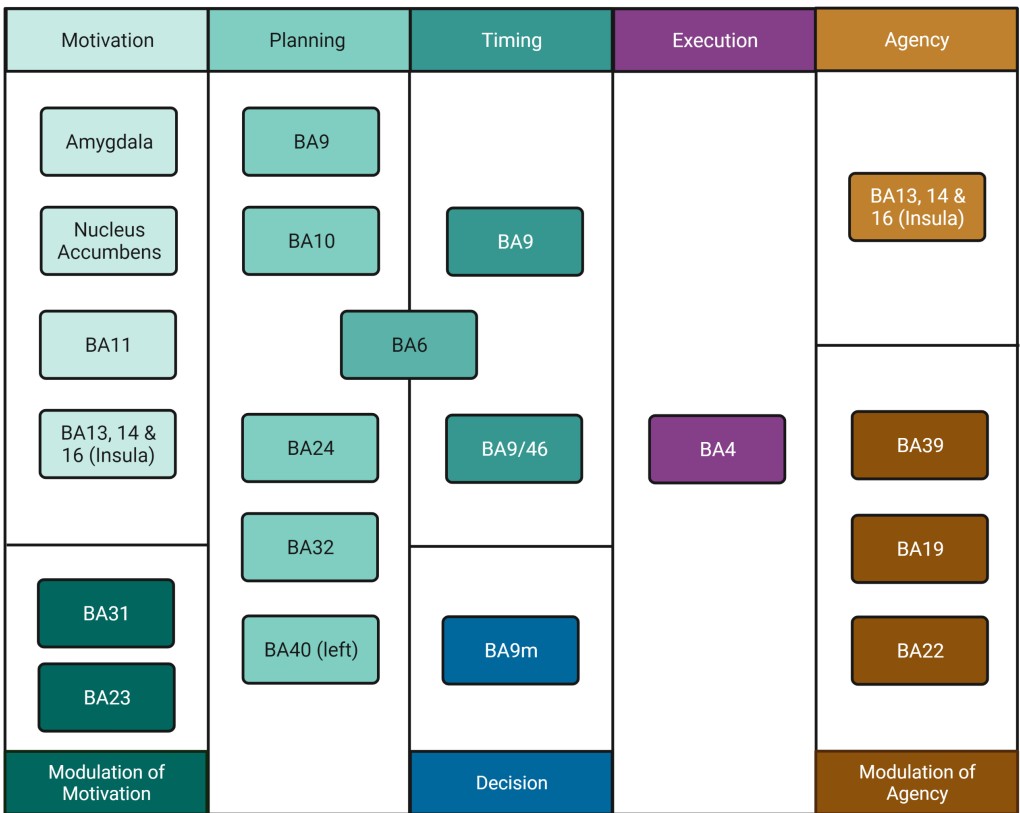

**Figure 1** **Process organisation for freedom of movement and areas involved.** The movement process includes Motivation in the amygdala, nucleus accumbens, BA11 and insula, a process which is modulated by areas BA31 and BA23. Planning involves areas BA9, BA10, BA24, BA32, left BA40 and BA6. Timing involves areas BA6, BA9 and BA 9/46, which may be inhibited by BA9m marking Decision, thus preventing movement. Execution involves area BA4. Perception of Agency involves areas BA13, BA14 and BA16 in the insula. If there is mismatch, inhibition of agency occurs in areas BA19, BA22 and BA39; if no mismatch is detected, BA4 delivers information to the insula and Agency arises. Created with and copyrighted by BioRender.com.

defined as real-valued functions on the vertices of a network, where the values provide a ranking which identifies the most important nodes. In particular, we computed the authorities, betweenness, degree, eigenvector, hubs, and closeness centrality measures. To further characterise the structure of the network, we calculated other parameters such as density (the ratio of the number of edges to the number of possible edges), network diameter (the shortest distance between the two most distant nodes in the network), global clustering coefficient (the probability that the adjacent vertices of a vertex in the network are connected), average of local clustering coefficients (how well connected are the neighbours of a vertex in a graph) and mean shortest path (average shortest path between any two nodes in a network). We also calculated assortativity, which is a measure of the preference of a node to attach to other similar nodes in terms of their degree.

We computed the modularity to measure the strength of division of a network into modules and used the Walktrap algorithm (*Pons & Latapy, 2005*) to identify communities

(also called groups or clusters). For the Walktrap algorithm, we included six steps implemented in igraph in R. The algorithm detects communities through a series of short random walks, implementing the idea that the vertices encountered on any given random walk are more likely to be within a community. The algorithm initially treats all nodes as communities of their own, then merges them into larger communities, and these into still larger, and so on. Essentially, it tries to find densely connected subgraphs (*i.e.*, communities) in a graph via random walks. The idea is that short random walks tend to stay in the same community.

The modules or communities of a network are subsets of nodes that are densely connected to other nodes in the same module but sparsely connected to nodes belonging to other communities. Since nodes within the same module are densely connected, the number of triangles in a modular network is larger than in a random graph of the same size and degree distribution, while the existence of a few links between nodes in different modules plays the role of topological shortcuts in the small-world topology. Systems characterized by this property tend to be described by small-world networks, hence exhibiting a high clustering coefficient and a short path length with respect to random networks.

To compute the small-worldness of the undirected volitional network, we consider this property and compute the mean local clustering coefficient $C$ of the network, the mean of the local clustering coefficients of one thousand randomly created networks $\langle C_r \rangle_{1000}$ with the same degree probability distribution function as the volitional network, the mean shortest path $L$ of the volitional network and the average of the mean shortest paths of the same one thousand random networks $\langle L_r \rangle_{1000}$. Many real-world networks have an average shortest path length comparable to those of a random network ($L \sim L_r$) and a clustering coefficient much higher than expected by random chance ($C \gg C_r$) (*Watts & Strogatz, 1998*). Watts and Strogatz proposed a novel graph model, called the Watts–Strogatz model, with a small average shortest path length $L$, and a large clustering coefficient $C$. We use this as the definition of a small-world network. Consequently, small-world networks are in between regular graphs with large $L$ and $C$ and random networks with small $L$ and $C$. To quantify small-worldness, we use the ratios (*Bassett & Bullmore, 2006*)

$$\mu = \frac{L}{\langle L_r \rangle_{1000}} \text{ and } \gamma = \frac{C}{\langle C_r \rangle_{1000}}$$

such that, for a small-world network

$$\sigma = \frac{\gamma}{\mu} > 1.$$

In this regard, $\sigma$ is the small-worldness measure and the higher it is from unity, the better the network displays the small-world property.

Finally, we visualised the entire connectivity of the volitional network and of each process in separate with connectograms using the Circos software (*Krzywinski et al., 2009*). Connectograms included centrality values which were normalised using the maximum value for each centrality measure, *i.e.*, taking each value and dividing it by the maximum number in each measure. This process set all centrality values from zero to one, making it easy for the software to create a gradient (a temperature graph) that matches the normalised value of each measure with colour intensity. Raw centrality values are provided in Table S1.

**Table 2 Results of network analysis.** First column, name of network characteristic including the small-worldness measure . Second column, the value or outcome of the corresponding calculation of the network characteristic in the first column.

| Characteristic | Values |
|---|---|
| Network type | Undirected |
| Adjacency matrix | Symmetric |
| Number of nodes | 82 |
| Number of edges | 2330 |
| Number of random networks used in the analysis | 1000 |
| Graph density | 0.350798 |
| Diameter | 4 |
| Global clustering coefficient | 0.6644389 |
| Average of local clustering coefficients | 0.6766388 |
| Mean shortest path | 1.74044 |
| Average of mean shortest paths of the 1000 random networks | 1.735682 |
| Absolute difference of the two values | 0.004757302 |
| Assortativity (r coefficient) | 0.2387355 |
| Modularity (Q coefficient) | 0.2031258 |
| Number of communities found | 4 |
| $\gamma =$ | 2.815384 |
| $\lambda =$ | 1.002741 |
| $\sigma (= \gamma / \lambda) =$ | 2.807689 |
| SCALE-FREE | 0 |
| SMALL WORLD | 1 |
| ASSORTATIVE | 1 |

## RESULTS

### Graph theoretical analysis reveals the volitional network is small-world

We calculated several network characteristics to assess the structure of the volitional network in its undirected form (Table 2). The entire volitional network consists of 82 nodes and the name of each node, according to the Brainnetome Atlas database, is presented in Table S1. Names are given in relation to Brodmann Areas they belong to. The undirected volitional network contains 2330 edges. Among those network characteristics in Table 2, we computed the graph density, diameter, global clustering coefficient, average of local clustering coefficients, mean shortest path, assortativity, modularity and small-worldness. The latter property is characterized by a relatively short minimum path length on average among all pairs of nodes in the network, and a high clustering coefficient. As we show in Table 2, $\sigma \cong 2.808 > 1$, hence the undirected volitional network is a small-world network. Finally, the undirected volitional network is assortative as the coefficient of assortativity $r$ is about 0.239, which is positive.

## Centrality measures suggest the nucleus accumbens as the most connected region

Once we measured general network properties, we calculated centrality measures of the undirected volitional network to identify important nodes (Fig. 2 and Table S1). We calculated centrality measures following the degree, hubs, betweenness, authorities, closeness, and eigenvector criteria (*Newman, 2010*) for each of the 82 nodes.

Following the degree centrality criterion, which calculates the number of edges a node has (*Newman, 2010*), nodes with the highest numbers of edges were left and right nucleus accumbens, right A23v, right A23d, right A6m, left A23v, left A6m, left A23d, left lateral amygdala, leftA31 and right lateral amygdala (Fig. 2A, Table S1). On the other hand, nodes with the least number of connections were the left cpSTS, right A39rv, right A39c, right A9/46d, right V5/MT+, left A39c, left A39rv, leftV5/MT+, left dorsal granular insula, left A40rd, right A40rd and right cpSTS (Fig. 2A, Table S1). These are mostly nodes involved in Modulation of Agency.

Betweenness centrality measures, which measure how important a node is to the shortest paths through the network (*Newman, 2010*), identified both left and right nucleus accumbens as the most important nodes, followed by the right A23v, right A23d, right and left vId/vIg, right A6m, right lateral amygdala, left A6m and left lateral amygdala (Fig. 2B, Table S1). Nodes with the lowest betweenness centrality values are left cpSTS, right A39rv, left dorsal granular insula, left A11l, right dorsal dysgranular insula, right A39c, right A32p, right A9/46d, left A32p and right hypergranular insula (Fig. 2B, Table S1).

With the closeness criterion, which indicates how close a node is to all other nodes (*Newman, 2010*), nodes with the highest values are the left and right nucleus accumbens, right A23v, right A23d, left A23v, right A6m, left A23d, left A6m, left A31 and left lateral amygdala (Fig. 2C, Table S1). On the other hand, nodes with the lowest closeness values are right A39rv, left cpSTS, right A40rd, left A40rd, right A39c, right V5/MT+, right A9/46d, right cpSTS, left A39rv and left dorsal granular insula (Fig. 2C, Table S1).

Hubs centrality measures, which calculate edges sent by each node (*Newman, 2010*), rank the left and right nucleus accumbens with the highest score, followed by the right and left A23d, right and left A23v, left and right A24rv, left A31 and left A9m (Fig. 2D, Table S1). On the other hand, nodes with the lowest hubs centrality values are the right A39rv, left cpSTS, right A40rd, right A39c, left A39rv, left A40rd, right cpSTS, right V5/MT+, right A9/46d and left A39c (Fig. 2D, Table S1).

Values with the authorities centrality criterion, which calculates edges received by each node, and with the eigenvector centrality criterion, which measures the influence a node has in a network (*Newman, 2010*), are identical to those of hubs centrality, and hence node ranking is also identical following these three criteria (Fig. 2D, Table S1). Together, these results show that the nucleus accumbens is one of the most connected regions in the volitional network as it presents the highest values according to all centrality criteria. Also, nodes involved in Modulation of Agency frequently have the lowest centrality values.
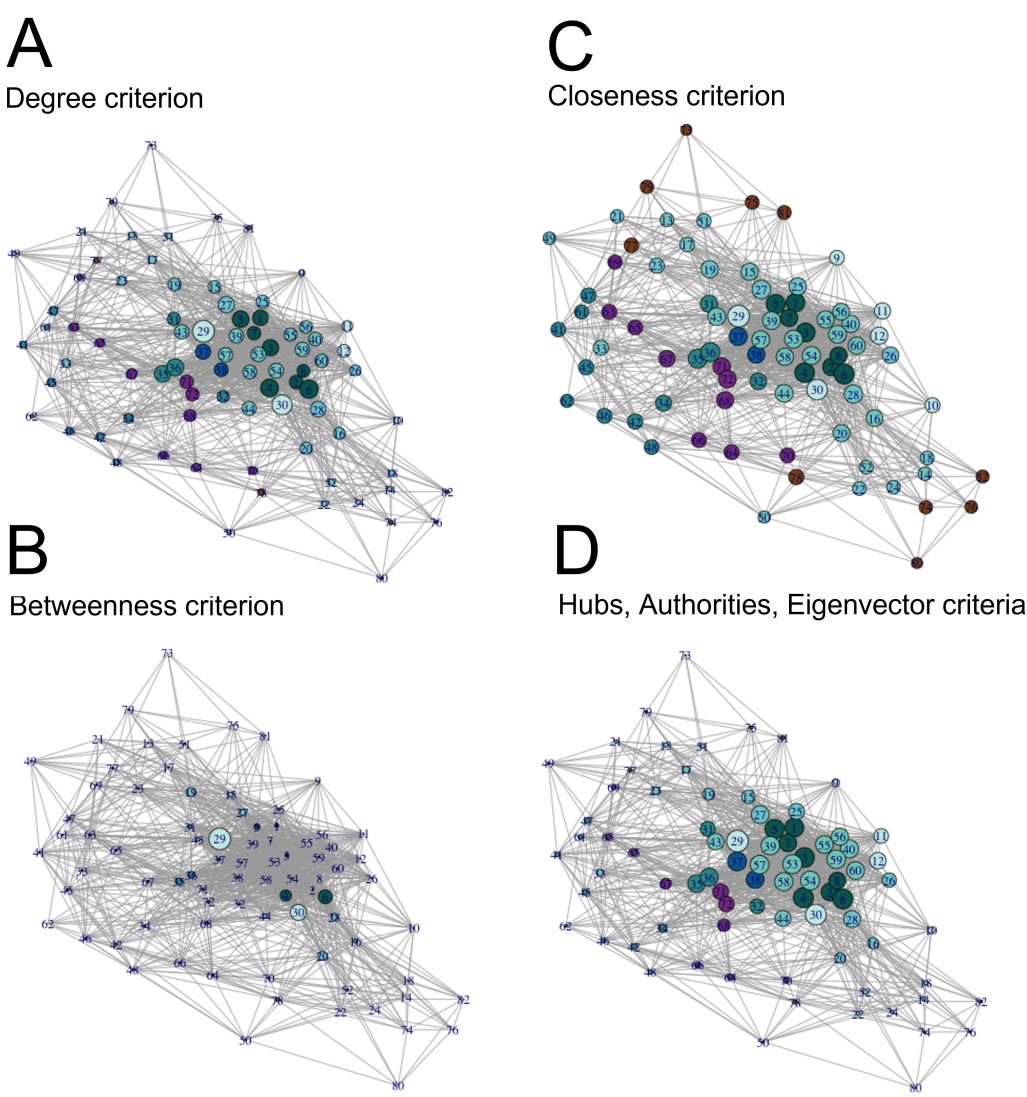

**Figure 2 Visualisation of centrality measures in the volitional network.** All 82 nodes are represented, their sizes proportional to each centrality measure in the (undirected) volitional network as indicated and colour coded as in Fig. 1. The code number and full name of each node are shown in Table S1. (A) Degree centrality criterion shows the left and right nucleus accumbens as nodes with most edges, followed by nodes involved in modulation of motivation, planning and timing. (B) Betweenness centrality criterion clearly shows the nucleus accumbens as the most important region to the shortest paths through the network, followed by sub-areas right A23v and right A23d. (C) Closeness centrality criterion shows that, in general, all nodes are close to each other. (D) Authorities, hubs and eigenvector centrality criteria show nodes involved in Motivation, Modulation of Motivation, Planning and Decision as nodes that receive and send more edges, as well as nodes possessing the most influential edges inside the network, respectively.

## Connectogram analysis shows a differential connectivity pattern along the movement process

Once we classified the entire movement process into processes and calculated structural properties of the volitional network, we inspected the connectivity of all areas involved

in each process and compared them with their centrality values by constructing connectograms using the Circos software (*Krzywinski et al., 2009*) (Fig. 3). This representation allowed the comparison of all 82 nodes in both hemispheres and the identification of bilateral differences. The entire connectivity of the structural volitional network is shown in Fig. 3A (see also Fig. S1).

Several brain regions are involved in Motivation (Table 1), and the connectogram representation reveals that nearly all of the 82 nodes are connected to at least another node in this process, with the exception of right A6vl, right A6cdl and right A9/46d (Fig. 3B, Fig. S2). Interestingly, most sub-areas involved in Modulation of Motivation show comparatively higher centrality values except for betweenness centrality (Fig. 3C, Fig. S3). The overall connectivity pattern of sub-areas involved in Motivation is extensive and similar in both hemispheres.

For Planning, most of the 82 nodes are connected to at least another node in this process, except right A39c in the parietal lobe and right V5/MT+ in the occipital lobe (Fig. 3D, Fig. S4). From all processes, Planning is the one that presents more connectivity. Connections involved in Timing occur in the frontal lobe mostly and some bilateral differences are observed (Fig. 3E, Fig. S5). For example, in the right insula, only one sub-area, right vld/vlg, is connected, whereas all six sub-areas conforming the insula are connected to at least another node in the left hemisphere (Fig. 3E, Fig. S5). Also, in the right parietal lobe fewer connections are present with respect to the left hemisphere and in the right occipital lobe (sub-area V5/MT+) no connection is observed as in the left hemisphere.

For Decision, connectivity is observed mainly between A9m and many sub-areas in the frontal lobe, parietal lobe, limbic areas, and subcortical nuclei in both hemispheres, with both ipsilateral and contralateral connections (Fig. 3F, Fig. S6). A bilateral difference is observed in the insula where half of the sub-areas in the left insula present connections, whereas none in the right insula do. Also, these three sub-areas in the left insula present the highest centrality values in that region. A similar situation occurs in the parietal lobe where only two sub-areas are connected in each hemisphere but they show the highest centrality values, with all four sub-areas reaching A9m.

For Execution, connections are observed among several sub-areas in the frontal and parietal lobes, in the insula, limbic areas and subcortical nuclei (Fig. 3G, Fig. S7). The overall pattern of connectivity looks symmetrical with slight differences such as in the left A11l and left A9l where no connection is observed as in the right hemisphere. In this process, no connections are observed in the temporal and occipital lobes.

Finally, for Agency, connectivity is extensive in all regions examined, and a bilateral difference is observed in contralateral connections between the left insula and the right frontal lobe, whereas those connections do not occur between the right insula and left frontal lobe (Fig. 3H, Fig. S8). Sub-areas presumably involved in Agency modulation (Fig. 3I, Fig. S9) also present bilateral differences. For example, connections exist between sub-areas in the left parietal and occipital lobes with limbic areas in the right hemisphere but the inverse is not observed.

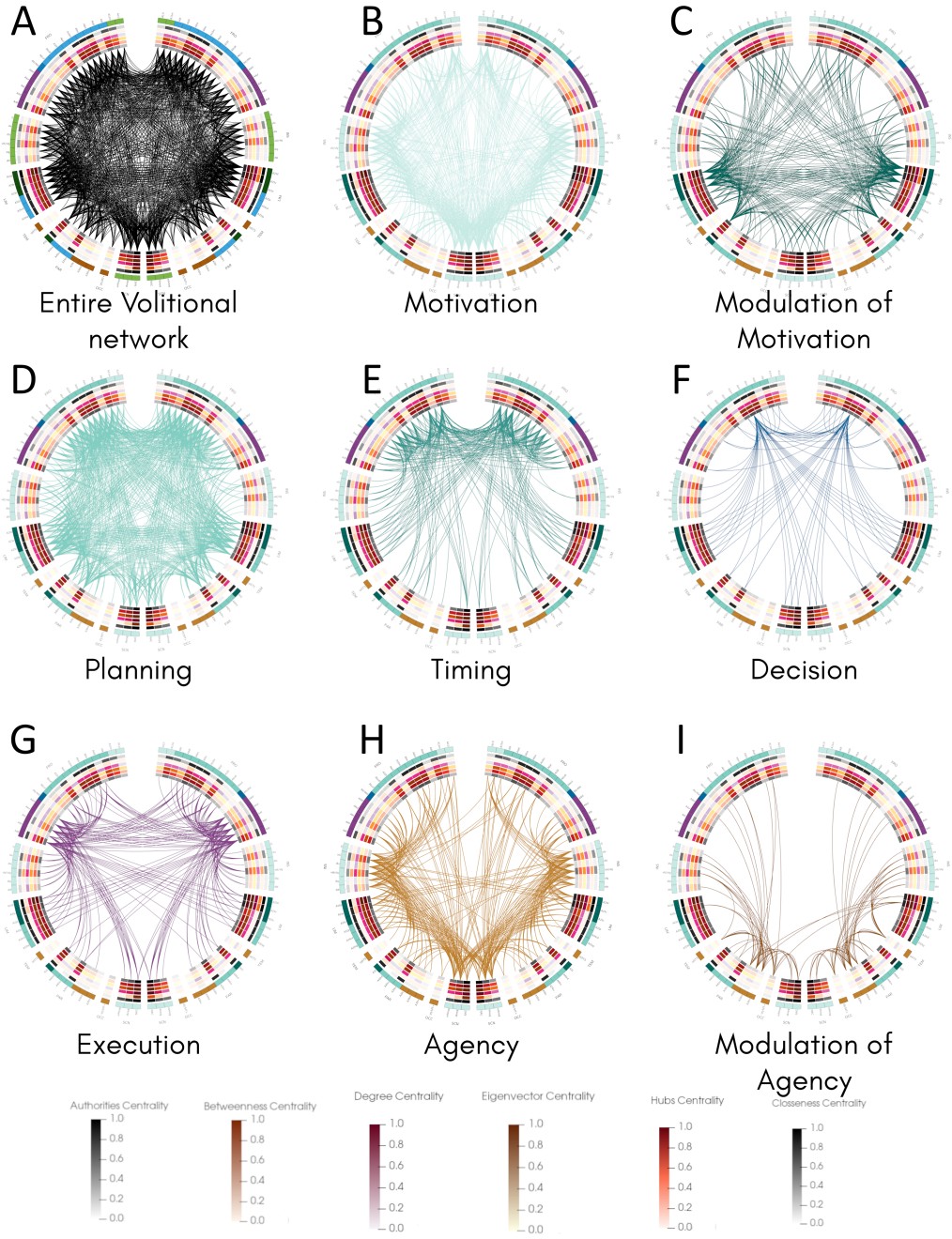

**Figure 3** **Connectogram analysis per movement process.** Circular representation of connections of each process made with Circos (*Krzywinski et al., 2009*). (A) Entire volitional network. (B) Motivation. (C) Modulation of Motivation. (D) Planning. (E) Timing. (F) Decision. (G) Execution. (H) Agency. (I) Modulation of Agency. All 82 nodes are represented at the periphery, grouped by brain regions. Nodes of the left hemisphere are shown at the left of the graph, and those of the right hemisphere at the right. Inside, concentric circles represent the six centrality measures (from outermost to innermost: authorities, betweenness, degree, eigenvector, hubs and closeness centrality criteria) and their values are colour coded following a temperature graph at the bottom. These values were normalised as described in Materials & Methods. Abbreviations: FRO, frontal lobe; INS, insula; LIM, limbic system; TEM, temporal lobe; PAR, parietal lobe; OCC, occipital lobe; SCN, sub-cortical nuclei.

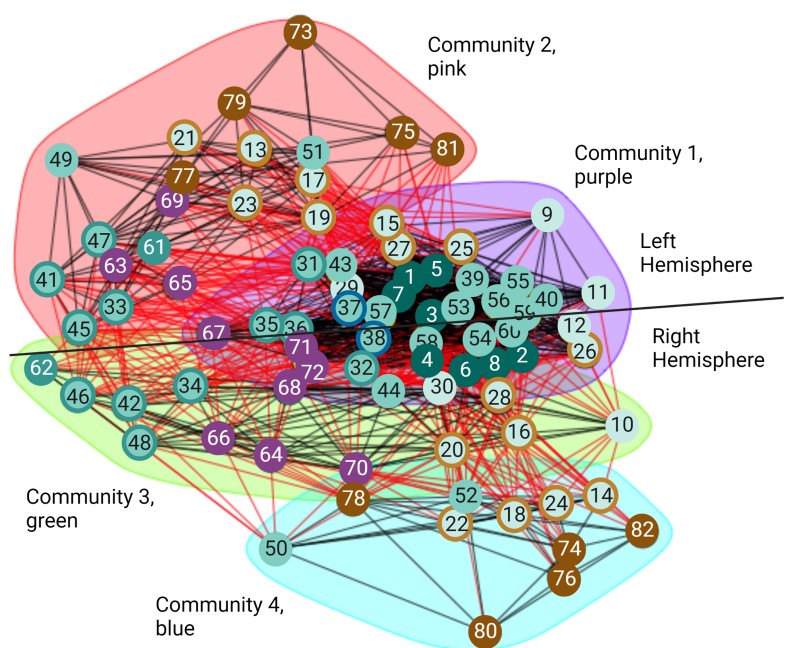

**Figure 4   Communities of the volitional network.** Results of community analysis by the Walktrap algorithm in the undirected volitional network. Four communities (communities 1–4) were found and most nodes from the right hemisphere are contained in three communities whereas those of the left hemisphere only in two. A line is drawn to separate nodes from both hemispheres. Code numbers of each node as in Table S1 and colour coded per process. Nodes with rings of a different colour represent nodes involved in two processes. Created with and copyrighted by BioRender.com.

Together, the pattern of connections among the 82 nodes of the structural volitional network as observed with connectograms shows a relatively extensive connectivity in all processes, with some bilateral differences. Processes with a comparatively reduced connectivity are Decision and Modulation of Agency.

## Community measures reveal the volitional network is more structured in the right hemisphere where agency is dominant

Even though small-worldness captures important aspects at the local and global scale of the structure in a network, it does not provide information about the intermediate scale. Properties of the intermediate scale can be better described by the community structure or modularity of the network (*Newman & Girvan, 2004*). In the undirected volitional network, we found it has the modularity value of 0.203 (Table 2). Even though this is a relatively small value, the entire undirected volitional network is modular. Hence, using the Walktrap method, we identified four communities (Fig. 4). The largest community (Fig. 4, community 1 shaded in purple) contains 34 nodes, which represents nearly half of all nodes in the undirected volitional network. This community contains nodes involved in all tasks except Modulation of Agency from both hemispheres. Both nodes involved in Decision (left and right sub-areas A9m) belong to this community.

Two adjacent communities (Fig. 4, community 2 shaded in pink, and community 3 shaded in green) contain fewer nodes. Interestingly, these communities contain nodes either from the left or from the right hemisphere, in contrast to community 1 that contains nodes from both hemispheres. The same happens with community 4 which only contains nodes from the right hemisphere.

A significant difference was found in communities containing nodes involved in Agency. In our community analysis, most nodes involved in Agency are in communities with fewer nodes, particularly those involved in Modulation of Agency highlighting their reduced number of connections also observed with connectograms and centrality values. However, the community that contains Agency nodes in the right hemisphere (Fig. 4, community 4 shaded in blue) is smaller than the corresponding one in the left hemisphere and does not contain Timing and Execution nodes as its left counterpart. This reveals an important difference between brain hemispheres in that the right hemisphere is more structured than the left with an additional community.

Taken together, these results reveal that the movement process initiates in nodes comprising a large community that later segregates into left and right hemispheres, with a subsequent segregation in an additional community present in the right hemisphere but not in the left. This might correlate with the fact that Agency is dominant in the right hemisphere (see also Discussion).

## DISCUSSION

We identified a structural volitional connectivity network divided into eight processes that span the movement process from Motivation to the sense of Agency at the sub-area level as characterised by the Brainnetome Atlas database. We identified 82 nodes from cortical and subcortical regions and used Graph Theory to calculate their network properties. Our results show that the undirected volitional network is small-world and contains four communities. However, nodes of the right hemisphere are contained in three of these communities whereas those of the left hemisphere only in two. Centrality measures indicate the nucleus accumbens is one of the most connected nodes in the network. The final structural network is consistent with a regulated mismatch mechanism for execution of movement and perception of agency.

### The undirected volitional network is small-world

Many studies have shown that the nervous system is small-world (*Hilgetag et al., 2000*; *Stephan et al., 2000*; *Sporns & Zwi, 2004*; *Eguíluz et al., 2005*; *Salvador et al., 2005*; *Achard et al., 2006*; *Humphries, Gurney & Prescott, 2006*; *She, Chen & Chan, 2016*; *Farahani, Karwowski & Lighthall, 2019*). Small-world networks are highly clustered (high clustering coefficient) as regular graphs or lattices but have characteristic small path lengths as random graphs; *i.e.*, they are between regular and completely random networks (*Watts & Strogatz, 1998*; *Lago-Fernández et al., 2000*). The undirected volitional network that we propose here falls into the category of small-world networks and adds yet another example of how the nervous system is organised in this manner to accomplish a particular process such as movement. The existence of long-range connections inside the volitional network between

different brain regions agrees with it being a small-world network providing shortcuts from otherwise highly separated nodes (*Watts & Strogatz, 1998*).

Small-world networks in the nervous system are reported to be both globally and locally efficient (*Latora & Marchiori, 2001*; *Vragović, Louis & Díaz-Guilera, 2005*; *Bassett & Bullmore, 2006*; *Achard & Bullmore, 2007*; *Bullmore & Sporns, 2012*), and so the structure of the volitional network is also compatible with it being an economical network. Dynamically, small-world networks have been shown to support synchronisation and coherent oscillations in certain conditions (*Lago-Fernández et al., 2000*; *Hong, Choi & Kim, 2002*; *Antonopoulos et al., 2015*). Small-world properties can also depend on the frequency used to measure the functional connectivity of the brain (*Stam, 2004*). The finding that the volitional network is small-world could set the ground for future studies of functional properties that make movement more efficient in health and make recovery from disease possible.

### Connectivity in the prefrontal cortex suggests a robust network for movement initiation

For most processes there are many nodes and edges (Fig. 3). Connections are particularly extensive in Motivation and Planning processes. This is understandable given that deciding to do something implies integrating different reasons (motivations), what to do from a vast number of options, and deciding how and when to do it, becoming demanding in the vastness of all possibilities. All these computations need a proper network for them to be resolved. In contrast, other processes such as Decision or Modulation of Agency present fewer connections, which may reflect their more direct nature. Decision only allows for a ''yes or no'' choice, and other processes do not have choices but are centred on the actual performance of movement and perception of agency. This also suggests that the less connected part of the volitional network is less resilient to damage, *i.e.*, damage to Decision or Modulation of Agency sub-areas may affect network performance irreparably. In contrast, damage in a random sub-area during Motivation or Planning may be less dramatic since other adjacent sub-areas could potentially compensate for it and maintain a certain degree of function of the entire network.

### The most connected nodes in the volitional network belong to ancestral areas

From all 82 nodes, nodes that ranked highest in all centrality measures were the left and right nucleus accumbens (Table S1). The nucleus accumbens is part of the ventral striatum and regulates motivational and emotional, as well as limbic-motor processes (*Salgado & Kaplitt, 2015*). The nucleus accumbens (or the striatum in general) is an ancient vertebrate structure (*O'Connell & Hofmann, 2011*; *Grillner & Robertson, 2016*; *Loonen & Ivanova, 2016*) and its evolutionary ancestry might be compatible with it being one of the most connected areas in our study. Nodes from BA23 ranked highest after the nucleus accumbens (Table S1). BA23 is part of the posterior cingulate cortex, an area which is generally involved in spatial processing and cognitive functions such as memory, attention and conscious awareness (*Leech & Sharp, 2014*; *Rolls, 2019*). The posterior cingulate cortex as such has been recognised in primates (*Armstrong et al., 1986*).

On the other hand, the less connected nodes were those belonging to BA39 (which modulates Agency) and BA40 (involved in Planning) in the parietal lobe and to the superior temporal sulcus (STS, which modulates Agency) in the temporal lobe. Parietal areas might be present in all mammals but this lobe has been further expanded in primates, particularly the posterior parietal cortex where BA39 and BA40 are found (*Kaas, Gharbawie & Stepniewska, 2011*; *Kaas, 2013*; *Kaas, Qi & Stepniewska, 2018*). The posterior parietal cortex has evolved in primates for complex grasping and tool handling towards the human lineage (*Almécija & Sherwood, 2017*; *Goldring & Krubitzer, 2020*), and the supramarginal gyrus in particular has been implicated in the processing of human advanced tool use (*Goldenberg & Spatt, 2009*; *Orban & Caruana, 2014*). Furthermore, the temporal lobe is unique to primates and, although the STS has been identified in different primate species, it has further evolved in humans for increased social abilities (*Ward et al., 2015*; *Bryant & Preuss, 2018*; *Patel, Sestieri & Corbetta, 2019*). Thus, at a broad scale, we observe an overall correlation between centrality values and evolutionary ancestry, with ancestral nodes presenting the highest centrality values while more recent brain areas involved in sophisticated human brain processes present the lowest. From a network perspective, this might imply that neurons in evolutionary more recent areas have had less time to make connections to other brain regions than neurons in more ancestral areas.

Our volitional network is a subset of the model presented by *Opris, Chang & Noga (2017)* in terms of a hierarchical network, although our model considers a more refined cortical parcellation. Our model includes only cortical and subcortical areas selected based on studies on volition, but *Opris, Chang & Noga (2017)* also included areas of the brainstem and spinal cord. We think that our volitional network could thus be extended to incorporate other regions from different parts of the central nervous system (cortical, subcortical, brainstem, and spinal cord) according to the model presented by *Opris, Chang & Noga (2017)*. We predict that it will still retain small-world properties. Also, the incorporation of more nodes and their centrality ranking could help to discern the establishment of nodes in a dynamical network across species if their ranking correlates with their evolutionary ancestry.

## Communities of the volitional network perform specific tasks during movement

For almost all the processes there is no obvious difference concerning which hemisphere of the brain presents more importance in this structural network except for perception of Agency, where we observed that nodes involved in Agency formed distinct communities in each hemisphere (Fig. 4). This could be reconciled with the fact that this process has been empirically found to be greater in the right hemisphere (*Ruby & Decety, 2001*; *Farrer & Frith, 2002*; *Farrer et al., 2003*; *Astafiev et al., 2004*; *David et al., 2007*; *Nahab et al., 2011*; *Chambon et al., 2013*; *Chambon, Moore & Haggard, 2015*), suggesting that a different network organisation might be necessary for Agency dominance in a given hemisphere. The analysis of our volitional network seems to imply that sub-areas involved in Modulation of Agency in the right hemisphere share common characteristics and create a distinct and smaller community (Fig. 4, blue shading), while sub-areas involved in Modulation of

Agency in the left hemisphere are found integrated with more nodes from other processes (Fig. 4, pink shading). From a structural network viewpoint, this differential grouping of the right hemisphere versus the left may be necessary to support the main role the right hemisphere has on Agency generation.

The community results are in accord with the general sense of movement and agency and its disorders. The large community 1 (Fig. 4, purple shading) is extensive with representation from all over the brain. This is what would be expected for the development of intention. For any movement, there will be input from external senses from the posterior part of the brain and from internal drives from the front part of the brain (*Rizzolatti, Luppino & Matelli, 1998*; *Rizzolatti & Luppino, 2001*). These will be integrated in frontal mesial areas such as the ACC and pre-SMA (*Paus, 2001*). Damage to some parts like the visual cortex will reduce visual driving of movement and in the hypothalamus might reduce driving from thirst; however, all other systems will drive normally. On the other hand, damage to integrative areas such as the ACC will lead to loss of driving from all causes and produce symptoms such as bradykinesia and abulia (*Fisher, 1984*; *Paus, 2001*). The bradykinesia in Parkinson's disease arises also from the decreased function of frontal mesial areas since a major output of the basal ganglia supports that region (*Hanakawa, Goldfine & Hallett, 2017*).

Communities 2 (Fig. 4, pink shading) and 3 (Fig. 4, green shading) can be thought of as being mirror networks in terms of their function and can compensate each other. For example, in right-handers, the left hemisphere is dominant for skilled movement for both right and left hands. Left parieto-premotor networks store skilled movements and are critical for their execution (*Rizzolatti, Luppino & Matelli, 1998*; *Rizzolatti & Luppino, 2001*). The right hemisphere can assist with movements of the non-dominant arm, and potentially can serve as a backup with left hemisphere damage, such as with stroke (*Chen, Cohen & Hallett, 2002*). This is analogous to the better-known situation of language which is also left hemisphere dominant. Damage just to parietal or premotor areas will lead to apraxia, loss of skilled motor actions, and when the primary motor area is also damaged, there will be hemiparesis (*Wheaton & Hallett, 2007*).

Community 4 (Fig. 4, blue shading), which only contains nodes from the right hemisphere, is more perceptual, including the feedforward-feedback matching that gives rise to the sense of agency. The right temporoparietal area is also dominant for body ownership, and damage can give rise to syndromes of neglect and anosognosia, unawareness of deficits (*Carota & Bogousslavsky, 2017*). Loss of agency, as seen for example, in functional movement disorders, is accompanied by dysfunction of the agency network (*Baizabal-Carvallo, Hallett & Jankovic, 2019*).

Finally, while our model provides a detailed description of the structural connectivity involved during the movement process, together with its network properties, given the characteristics of the data, results drawn from this study remain as a set of hypotheses with intrinsic limitations. Also, information about the "intensity" of connection between sub-areas was not accounted for, together with axon-related characteristics such as myelination patterns that could affect functionality. Given that our study is of network structure, more studies should be done with resting state and task-related fMRI to explore the function of

the volitional network more directly. Especially important is performing more MRI, EEG, and MEG studies to obtain the connectivity of brain areas that were excluded from our analysis and, therefore, this volitional network might be incomplete. Still, we hope that our study constitutes a starting point for a more detailed description of the volitional network from a structural perspective.

## CONCLUSIONS

Our research identifies a structural connectivity brain network for decision of movement from a connectomics approach. We identified an undirected volitional network with small-world properties and a differential community structure between hemispheres for nodes involved in Agency. We suggest that the most critical sub-areas are the ones ranking higher with respect to their centrality values, and that these should be studied in more detail in experimental settings. This model expands and refines the cerebral connectivity for movement realisation more than previous models. Our volitional network could serve as a starting point for the construction of more refined models for movement and could potentially help in the understanding of resilience to brain damage. This type of analysis could also be used to explore the importance of specific sub-areas for other processes such as consciousness or neuro-psychiatric conditions by identifying their specific structural connectivity networks.

## ACKNOWLEDGEMENTS

We thank Ana V. Arredondo Robles, Gabriel Santos Vázquez, Claudia Azuela Peña and Juana I. Villareal Valdés for their excellent insights about the medical, philosophical and mathematical observations.

### Funding
Mark Hallett is supported by the NINDS Intramural Program. The funders had no role in study design, data collection and analysis, decision to publish, or preparation of the manuscript.

### Grant Disclosures
The following grant information was disclosed by the authors:
NINDS Intramural Program.

### Competing Interests
The authors declare there are no competing interests.

### Author Contributions
- Diego Alonzo Rodríguez-Méndez conceived and designed the experiments, performed the experiments, analyzed the data, prepared figures and/or tables, authored or reviewed drafts of the article, and approved the final draft.

- Daniel San-Juan analyzed the data, authored or reviewed drafts of the article, and approved the final draft.
- Mark Hallett analyzed the data, authored or reviewed drafts of the article, and approved the final draft.
- Chris G. Antonopoulos performed the experiments, analyzed the data, prepared figures and/or tables, authored or reviewed drafts of the article, and approved the final draft.
- Erick López-Reynoso performed the experiments, analyzed the data, prepared figures and/or tables, authored or reviewed drafts of the article, and approved the final draft.
- Ricardo Lara-Ramírez conceived and designed the experiments, analyzed the data, prepared figures and/or tables, authored or reviewed drafts of the article, and approved the final draft.

### Data Availability

The raw values of the six centrality measures we calculated and the codes and full names of all 82 nodes are available in the Supplementary Files.

### Supplemental Information

Supplemental information for this article can be found online at http://dx.doi.org/10.7717/peerj.13602#supplemental-information.

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
