# Peer review of "A new model for freedom of movement using connectomic analysis"

_PeerJ, doi:10.7717/peerj.13602_

## Round 0.1 · original submission · Major Revisions

As you can see, both reviewers felt that the manuscript was interesting and well written. However, both also expressed serious reservations about the validity of the model, the identification and parcellation of Brodmann's areas as hubs in the connectivity network, and the attribution of causality across network connections. I believe that these concerns are addressable, but substantial revision and reanalysis of the proposed model will be required. I urge you to consider carefully the specific comments of the reviewers, and to consider submitting a thoroughly revised manuscript for further review. Thank you for submitting your work to PeerJ, and I look forward to reading the next version of this work.

On a separate note, I observed that the reporting of numbers with respect to edges and connections throughout the paper is somewhat inconsistent, with many percentages reported to 2 decimal places, with no referent for how many actual connections were described (e.g. "78.96% of edges"). This implies a degree of precision to one part in 10000, which is both artificially high, and also inconsistent with the reporting of numbers elsewhere in the manuscript. In the revised version, please attend carefully to the reporting of significant figures.

Reviewer 1 ·

Basic reporting

The manuscript is in general well-written and easy to follow. The tables and figures are thoroughly described, and I thought Figure 4 was well-designed.

Experimental design

The authors adopt a causal view of volitional action as a sequence of steps. Then they identify brain regions associated with these functions from the BrainMap database of prior fMRI studies. Next, they try to perform an abstract “tract tracing” through these functionally defined regions using “tracts” identified from the Brainnetome database with a graph-theoretic interpretation. This network is then interpreted as the key network for voluntary action.

In a very general sense, I see the merits of the author’s intentions. Prior imaging studies with fMRI often focus on describing a cognitive ability as a functional network, which is effectively a list of brain regions that are engaged in tasks involving that cognitive ability. However, relating these regions to actual structural connectivity constraints is an important next step, especially using the new publicly available connectome databases. Although this is an important goal, in my evaluation, the author’s execution of this goal seems a bit too preliminary and overly simplistic in its current form.

MAJOR CONCERN 1
In general, the issue of how all the pieces fit together for voluntary action is important. However, I’m having difficulties in determining whether this manuscript is a connectome-oriented (i) review, or (ii) quantitative meta-analysis, or neither.

Prior reviews on this topic have sought to provide integrative overviews for voluntary action. Their objective is often to provide a conceptual scheme to integrate heterogeneous considerations and findings. Other studies, for example with Activation Likelihood Estimation (ALE) seek to quantitatively integrate the heterogeneous findings from different experimental studies to identify commonalities and, importantly, to statistically assign a confidence to these commonalities.

The current study does not seem to be a quantitative meta-analysis. [See, for example, Zapparoli, L., Seghezzi, S., & Paulesu, E. (2017). The what, the when, and the whether of intentional action in the brain: a meta-analytical review. Frontiers in human neuroscience, 11, 238.]. In Table 1, the evidence for each “module” and the associated brain regions is based on isolated studies and there is per se no analysis of whether empirical studies of voluntary action are consistent with these regions. The assigning of function to brain activity is based on associations from studies on other topics obtained from Brainmap rather then volition itself. For example, consider BA11’s role in "motivation". Might BA11 be broadly associated with emotion and cognition? Yes, possibly based on studies on different topics. However, is BA11’s role in voluntary action related to motivation? This is a far more complex, ambiguous answer and is not addressed.

In summary, there is little evidence provided for the reliability of the different regions and their functions listed in Table 1, in studies specifically about voluntary action. This is an issue that needs to addressed by the authors. Are the regions listed in Table 1 (1) identified based on broad conceptual considerations of general function, or (2) from quantitative regularities observed in studies explicitly about voluntary action? Without this clarification, the regions listed seem quite ad hoc.

MAJOR CONCERN 2
The authors describe a causal model in Figure 1 which is central to the entire study. In a logical sense, the steps are reasonable (motivation, what/how/whether/when to move, etc.). They are decisions that must necessarily be involved at some point before an action. However, the entire study is based on a strict interpretation of the order in which these steps occur (Figure 2). This is an extraordinary simplistic assumption and one that is indefensible without context.

To provide a simple example, I can decide today that I will move one of my fingers at 10am tomorrow (when) and postpone the decisions about which finger to move, by how much and whether to do actually do it until the time arrives. This is a challenging aspect in Libet’s studies.

The more serious concern is the extent to which brain networks could be assigned to these causal functions. For a visually guided action, classical psychological models suggest that there is an ordering of events. [seen]=>[identified]=>[response selected]=>[executed]. These are logically necessary. However, their NEURAL implementation and sequencing poses an entirely different issue. A minimum requirement is that the stimulus be sensed and the action be executed. So, information has to putatively travel from the visual cortex until the motor cortex where signals to execute the action can be discharged through the cortico-spinal tract to the effector. However, the ordering of these steps in the brain and whether they actually occur in such an order has been anything but controversial. For instance, it does not preclude that the responses are prepared BEFORE the stimulus arrives.
• Cisek, P., & Kalaska, J. F. (2010). Neural mechanisms for interacting with a world full of action choices. Annual review of neuroscience, 33, 269-298.
• Ploran, E. J., Nelson, S. M., Velanova, K., Donaldson, D. I., Petersen, S. E., & Wheeler, M. E. (2007). Evidence accumulation and the moment of recognition: Dissociating perceptual recognition processes using fMRI. The Journal of Neuroscience: The Official Journal of the Society for Neu- roscience, 27(44), 11912–11924. https://doi.org/10.1523/ JNEUROSCI.3522-07.2007
• Viswanathan, S., Abdollahi, R. O., Wang, B. A., Grefkes, C., Fink, G. R., & Daun, S. (2020). A response‐locking protocol to boost sensitivity for fMRI‐based neurochronometry. Human brain mapping, 41(12), 3420-3438.

In summary, the strict ordering assumed by the authors is highly problematic. It would be important to assess the extent to which the results rely on this sequential progression. How similar would the results be if the ordering of the steps were different? Also, why is the assumption of sequential ordering even necessary when considering the structural connectivity? If no assumptions about ordering are made, how much would the conclusions about connectivity differ? For instance in Figure 3, if the connectivity matrix is considered without restrictions on the steps then there would be a much higher number of edges. How much does this effect the interpretation?


MISCELLANEOUS
There are several ambiguities in discussing connectivity. From Lines 127-137, it is unclear whether the connectivity is based on structural or functional connectivity. What if functional connectivity were to be considered for the same regions? Why would this not be a suitable view since voluntary action is also a functional pattern of neural activity

How do these findings relate to the connectivity conclusions of this work → Zapparoli, L., Seghezzi, S., Scifo, P., Zerbi, A., Banfi, G., Tettamanti, M., & Paulesu, E. (2018). Dissecting the neurofunctional bases of intentional action. Proceedings of the National Academy of Sciences, 115(28), 7440-7445.

The authors might want to consider looking into the Brain Connectivity and Behavior toolbox (Michel de Schotten) [http://toolkit.bcblab.com/] that has sought to do what the authors are doing for lesion analysis. Specifically, given a lesion, the toolbox provides a means to evaluate the tract passing through that lesion to identify cortical regions connected to that lesion.
• de Schotten, M. T., Foulon, C., & Nachev, P. (2020). Brain disconnections link structural connectivity with function and behaviour. Nature communications, 11(1), 1-8.
• Nozais, V., Forkel, S. J., Foulon, C., Petit, L., & de Schotten, M. T. (2021). Functionnectome: a framework to analyse the contribution of brain circuits to fMRI. bioRxiv, 2021-01.

Validity of the findings

With the issues listed above, the validity of the findings are low and the interpretation in the Discussion considerable exceeds what might be reasonably concluded from the findings.

Reviewer 2 ·

Basic reporting

Major
1. Much of the manuscript is written clearly with appropriate references throughout. In the Introduction, the description of Hallett’s neuroanatomical model of volition (Lines 45 - 80) could be reorganized to improve clarity and remove redundancy. It would also be helpful to describe some of the empirical evidence for the model (i.e., why is pre-SMA implicated in movement timing?)
2. The labeling of Brodmann areas are sometimes inaccurate, which raises concerns about the interpretation of results. For example, the authors equate BA6 with pre-SMA, but BA6 more generally encompasses SMA and premotor cortex in addition to pre-SMA. Likewise, it is not clear that BA14 is part of the insula in humans. In general, the use of Brodmann’s areas in the text is sometimes confusing—I think it would be easier to conceptualize the model if the authors used appropriate anatomical labels along with the description of the BAs or sub-areas (ex. Line 206 could be modified to: “since BA9m (dFMC) has been implicated in self-initiated (conscious) inhibition of movement”)

Minor
1. The phrase “idea of liberty” seems out of place here, as the conceptual model is specific to the study of action, and it is unclear if it generalizes to other cognitive domains.
2. There are some awkward sentences and grammar (e.g., Lines 40-41 “We feel capable of deciding our actions and execute (or stop) them”; Line 31 “less connections”) and colloquial language (“we checked”/“we wondered”).
3. Why does the step organization start at Step “0”? In addition, could the Steps with an inhibitory component be labeled “a” and “b” (e.g., Step 5a?)
4. Fig. 2: Add legend for color coding the different steps.
5. Fig. 3: This figure is low resolution, which made it difficult to discern region names

Experimental design

Major
1. The research question and identified knowledge gap could be more strongly motivated. As currently written, it is not obvious how the present study extends previous neuroanatomical models of volition other than mapping out the network in more detail. Instead, a stronger emphasis could be placed on the utility of a connectomic approach for (a) confirming that Hallett (2016) model is plausible, in terms of human structural connectivity, and (b) identifying smaller sub-regions within the broader volitional network that may serve as critical nodes to investigate in future studies.
2. If the goal was to perform a connectomic analysis, I do not understand why the authors chose to simply “count” edges rather than use graph theoretic measures well-described in the literature (Bullmore and Sporns, 2009). The authors make inferences about complex networks concepts, such as “hubs”, “rich club”, and “robustness”, but these can be actually quantified by assessing node degree, clustering coefficient, centrality, etc. With the current approach, these designations about network properties seem only speculative.
3. I am struggling to follow the logic of the network analysis, as detailed in the Results, Fig. 3, and Table 2. First, many of the results seem to be a direct consequence of the regions selected for each step, rather than revealing anything in particular about the properties of the volitional network. For example, “For the inhibitory component of step 0, the dorsoposterior cingulate cortex (BA23) is the most connected node with 79.28% of total bilateral edges, leaving the ventroposterior cingulate cortex (BA31) with only 20.72%” (Lines 174-176). But this difference can be explained by the fact that six BA23 subregions were included (right and left BA23d, v, and c) that have edges with each other, relative to only two BA31 subregions. Of greater concern, the authors state that “The insula module had 100% of edges arriving from nodes active at step 4, suggesting it is the only area capable of generating positive agency” (Lines 232-234). This results seems to stem from the decision to only include insular cortex subregions in that step’s analysis. If this is the case, how can the edge contribution of the insula be anything other than 100%? The argument appears to be circular.
4. Lines 130-137: There needs to be a more complete justification for the selection of regions from the Brainnatome atlas. How exactly did the authors select the regions to be included in each step?
5. The connections between regions in Figs. 1 & 4 are very nicely visualized (especially Fig. 4). However, it should be more clear which aspects of the model are assumptions from previous work vs. results from the present analysis. For example, the Figures show inhibitory and feedforward connections between regions, but there is no way to know this from the Brainnatome connectograms alone.

Minor
1. Fig. 3: Label whether each sub-region is right or left hemisphere. Also, it does not make sense to color the matrix elements outside the black boxes blue, as this would imply that there is no edge between the regions (even though connections outside the black boxes were not assessed). In general, consider a different visualization approach for Figure 3. It might be helpful to try a circle plot visualization (Irimia et al., 2012) separately for the different steps.
2. What is meant by “this area is not present in the right hemisphere”? (Line 182) and “lack of BA39 in the left hemisphere” (Line 247). Please clarify.
3. It is unclear what BrainMap adds to the analysis—why is it important to list the various functions of a particular region if it is not relevant to volitional control?

Validity of the findings

Major
1. Related to the comment above, I find the conclusions about the percentage of edges (e.g., “Indeed, 78.96% of edges happen during the first three steps.” Line 286) problematic given that they could be mainly driven by the number of regions selected for the different model steps. The interpretation is interesting (delineation of early flexible motivation and selection processes vs. later simpler initiation and agency processes), but not convincingly supported by the results.
2. Likewise, the discussion of implications of the model for neurological disorders is interesting but beyond the scope of the study, especially since similar ideas were discussed in previous work (Kranick & Hallett, 2013). These sections could be removed or condensed significantly.
3. One of the central claims of the study is that “From the structure of this model, we propose that consciousness is not required for movement generation” (Abstract, 33-34). I do not really understand this point and I am not sure how the presence or absence of conscious processing can be determined from the model.

Additional comments

The overall idea and motivation for the study is interesting and could have important implications for future studies on the volitional control of action and sense of agency. My main concern is the emphasis on utilizing a connectomic approach without taking advantage of established measures for graph analysis of brain connectivity data sets.

---

## Round 0.2 · Major Revisions

As you will note, both reviewers indicated that the revisions to the previous version were a substantive and crucial improvement to the manuscript. However, the remaining issues raised by Reviewer 2 need to be resolved prior to consideration for publication. In light of Reviewer 2's comments, I believe that the remaining issues can be addressed without substantial new analysis or the need to collect additional experimental data. In your reply and revision, please identify the specific changes and your responses to the reviewers' comments, and I look forward to receiving your revised manuscript. Thank you again for choosing to submit your work to PeerJ.

Reviewer 1 ·

Basic reporting

I would like to commend the authors for the considerable amount of work done to revise the manuscript. I think this revision has vastly improved the manuscript’s content, rigor and clarity. All my major concerns have been sufficiently addressed.

Supplementary Table 1 is important but was missing from the review materials.

A minor suggestion is that the labeling of the figures and the captions could use improvement as they contain a lot of important information.

Figure 2: The caption is a bit too sparse to describe the figure.
• The panels 2E and 2F do not seem to be referenced anywhere in the text
• It could be useful if the colors of the nodes in the graph correspond to the colors in Figure 1 (as is nicely done in Figure 4) to relate these measures to the different "tasks".

Figure 3 is interesting and important but it was not possible to see any details even when zooming into the pdf. This seems to defeat the purpose of displaying the figure. Maybe the authors could consider providing high-resolution vector graphics images of each of these connectograms in the Supplementary Material so that the labels and values could be examined by readers. The authors could add the name of the task (e.g., “Motivation”, “Agency”, etc.,) next to the corresponding connectogram to make it easy to understand this figure.

Figure 4 – a legend with the color and task could be added to the figure so that the reader does not have to keep referring back to Figure 1 to find out which task corresponds to each color.

Figure 5 seems relevant after Figure 1 rather than helping to organize the information from the quantitative graph theory analysis. A suggestion could be to move it a position after Figure 1, and before the numerical results.

Experimental design

no comment

Validity of the findings

no comment

Reviewer 2 ·

Basic reporting

The revised manuscript is well-written, logically organized, and much improved.

Minor comments
1. Some of the details of the network analysis presented in the Results should be moved to the Methods (e.g., computation of small-worldness, lines 225-240)
2. I am not sure "Tasks" is the best word to use for the model description. Perhaps "Processes" is more appropriate? Additionally, referring to 6 tasks throughout the manuscript is a bit confusing, as sometimes 8 separate results are presented (e.g., the 6 tasks + 2 task modulators).
3. I preferred the naming scheme of "How", "When", and "Whether" (or similar) to "Planning", "Timing", and "Decision." The latter, introduced in this revision, typically encompass a broader set of processes than implied in the manuscript (e.g., "Decision" is commonly used to refer to selection of a particular action goal, effector, etc. rather than the go/no-go decision described in the Introduction).
4. It would be good to reinforce that the analysis is based on structural, not functional, connectivity throughout the manuscript (one example, line 138 "Here, we identify the *structural* connectivity network")
5. Figure 2 is visualized nicely; however it is difficult to appreciate without referencing the Supplemental Table (also true for Fig. 4 to a lesser extent). At the least, it would be good to color code the nodes to its corresponding Task, as is done with the other Figures. Additionally, since 2E and 2F are redundant with 2D, they should be removed.
6. There are a few instances where the old "Steps" terminology is used (e.g., Lines 478, 500, 508-510).

Experimental design

I appreciate the authors' extensive revisions and new graph theoretic analyses. The approach is much improved and addresses many of my previous concerns. However, some concerns with the selection of regions remain, along with additional concerns with the new results.

1. The results of the connectivity analysis obviously rests on the selection of regions. Although the Introduction now provides more information supporting their choices, I still find the region selection is somewhat arbitrary and lacks strong justification, for reasons detailed below:
a. The selected regions do not appear to fully align with Hallett's model, missing regions in some cases (e.g., hypothalamus, non-limbic basal ganglia) and adding regions in other cases (e.g., insula for Agency--my understanding is the Hallett model mainly identifies parietal regions, and that the sense of agency arises from parietal-premotor connections).
b. Though it is reasonable to include regions that extend Hallett's model based on other literature, the the decision to include additional regions is inconsistent. For example, the authors cite a study showing posterior cingulate activation during extrinsic motivation to act (Lee et al., 2012). In a follow-up study (Lee & Reeve, 2013), the angular gyrus was activated in extrinsically motivated contexts, yet this region is not considered for Motivation. Similarly, Nahab et al. (2011) found many regions, including supramarginal gyrus, precuneus, dlPFC, and cerebellum implicated in Agency, in addition to the superior temporal gyrus referenced and included in the model.
c. I am curious as to why the authors did not constrain their regions for the Planning, Timing, and Decision Tasks from meta-analyses of volitional action (e.g., Zapparoli et al., 2017, as mentioned previously by Reviewer 1).

Overall, I do not necessarily think the analyses need to re-done with new regions, but there needs to be a much stronger rationale for the approach to region selection. The exclusion of the classically "motor" regions of the basal ganglia (e.g., dorsal striatum) is particularly concerning, given that it may share a similar connectivity profile to the nucleus accumbens, which was identified as a principle node in the network in the Results.

2. All BA6 sub-regions are included in the Planning and Timing tasks, which is reasonable, however, as mentioned in my previous review, these regions extend beyond Pre-SMA. It is better characterized as Pre-SMA/SMA/Premotor cortex, and any Results with BA6 should reflect potential contributions from each of these regions rather than just Pre-SMA.

3. Some of the selected regions were restricted to one hemisphere (e.g., right BA39), yet in the model all regions were included bilaterally. Why is this the case, if left BA39 for example, is not considered to be part of the volitional network? How does this influence the hemispheric differences found in the connectivity analysis?

4. Figure 5 seems out-of-place in this revision, as the visualization seems to be mainly based on previous literature rather than incorporate connectivity findings from the present results. For example, BA4 is shown to have connections to the insula and BA6, yet the Results for Execution (Fig. 3G, Lines 316-17) show many more connections between BA4 and other areas. The nucleus accumbens, which features prominently in the Results, is not included. The figure may also be misleading in its depiction of a sequence of steps, rather than reflecting the undirected analysis performed.


Minor
1. Why is the amygdala attributed to Agency (in addition to Motivation) in Figure 4 and the supplemental Table?
2. Much of Discussion section 3.4 (Line 476) is no longer supported by the current Results. In particular, the statement "We found that without the insula the entire mismatch system becomes ill-connected since, with its absence, other areas would be rendered isolated" refers to the previous version of the manuscript.
3. The insula is not discussed with respect to Agency until the Discussion (Line 479). Please provide rationale in the Introduction.
4. Line 157. Do you mean "Areas thus identified are at the level of Brodmann areas or *subcortical areas*"?
5. Could the authors provide a reference for that notion that a network is small-world if Sigma > 1?

Validity of the findings

The validity of the findings would be improved with a much stronger rationale for region selection for the different Tasks of the volitional model.

---

## Round 0.3 · accepted · Accept

Thank you again for your submission, and for your careful attention to the comments and questions raised by the reviewers. I believe this process has resulted in a very high-quality final manuscript, and I look forward to seeing the final published version.